# SCALING REASONING, LOSING CONTROL: EVALUATING INSTRUCTION FOLLOWING IN LARGE REASONING MODELS

## ABSTRACT

Instruction-following is essential for aligning large language models (LLMs) with user intent. Yet recent reasoning-oriented models, despite their strong performance on complex mathematical problems, often fail to comply with simple natural language directives. In this work, we analyze the interaction between reasoning ability and instruction adherence in large reasoning models (LRMs). Using a controlled evaluation framework (MathIF), we uncover a persistent trade-off: as models scale reasoning capacity through long chains-of-thought or reinforcement learning on reasoning traces, their obedience to instructions degrades, particularly when generation length grows. We further show that interventions such as constraining or repeating instructions can partially restore compliance, but typically at the expense of reasoning performance. Taken together, our findings expose an intelligence–obedience dilemma in current training paradigms and underscore the need for instruction-aware approaches to developing controllable reasoning models.

## 1 INTRODUCTION

Recent advancements in Large Reasoning Models (LRMs) (Qu et al., 2025), such as o3 and o4-mini (OpenAI), DeepSeek-R1 (DeepSeek-AI, 2025), and K1.5 (Team et al., 2025), have demonstrated impressive capabilities in mathematical reasoning, including solving olympiad-level problems (He et al., 2024a; Hendrycks et al., 2021; Veeraboina, 2023) and automating formal theorem proving (Ren et al., 2025). These breakthroughs have sparked growing interest in scaling chain-of-thought (CoT) reasoning (Wei et al., 2022), where models produce explicit multi-step explanations to solve complex tasks. Typical approaches include imitation learning, e.g., supervised fine-tuning (SFT), and reinforcement learning with verifiable rewards (Su et al., 2025), both of which aim to strengthen model intelligence across various tasks and scales.

Despite these advances, instruction following, i.e., the ability to accurately and reliably comply with user directives, has received comparatively little attention in the context of LRMs. Yet this ability is critical for real-world alignment and safety (Gu et al., 2025). Our empirical evaluations on IFEval (Zhou et al., 2023) and FollowBench (Jiang et al., 2023) reveal a consistent pattern: although LRMs excel at mathematical reasoning, they often fail to follow even simple instructions. This raises an important question: ***As reasoning scales, do models become more intelligent yet less controllable?*** Unfortunately, existing instruction-following benchmarks are ill-suited for answering this question. Most are designed for general-purpose language models and lack coverage of math-specific reasoning behaviors. In contrast, LRMs are typically trained on math-heavy datasets and optimized specifically for problem-solving capacity. This gap highlights the urgent need to evaluate whether increasing intelligence in specialized reasoning models inherently leads to diminishing control over their behavior, an issue at the heart of instruction alignment for advanced LRMs.

To probe this phenomenon, we design **MathIF**, a controlled evaluation framework tailored for mathematical reasoning. Rather than serving as an end in itself, MathIF provides a systematic way to stress-test obedience within the mathematical reasoning domain. It combines 15 Python-verifiable constraints across four categories into compositional queries, and embeds them within math problems spanning diverse difficulty levels. Applying this setup to 25 recent LRMs, we uncover three consistent

findings: (1) instruction-following fidelity remains strikingly low across scales and architectures, with even the strongest open model (Qwen3-14B) achieving only 50.71% strict compliance; (2) obedience further deteriorates as task difficulty or constraint complexity increases; and (3) model size alone does not predict controllability. These results highlight a fundamental tension between reasoning strength and instruction adherence that persists across today's state-of-the-art LRMs.

Our analysis uncovers a persistent interference between instruction-following and reasoning capabilities, manifesting at both training and inference stages. Reasoning-oriented strategies such as supervised fine-tuning and reinforcement learning reliably strengthen mathematical problem-solving, yet simultaneously degrade adherence to user instructions. This degradation becomes especially pronounced as chain-of-thought (CoT) length increases, since longer reasoning paths widen the contextual distance between the original directive and the final answer, making faithful execution more difficult. Conversely, enforcing brevity by limiting CoT length improves instruction-following performance, but at the cost of reasoning depth and accuracy.

Taken together, these findings reveal a consistent pattern: ***gains in reasoning ability often come at the expense of controllability***. This trade-off poses a central challenge for LRM development: optimizing purely for intelligence can undermine alignment, and future training paradigms must reconcile the tension between capability and obedience. Building on this perspective, our contributions are three-fold:

• We design MathIF, a controlled evaluation framework tailored to probing instruction adherence in mathematical reasoning tasks.
• Through a large-scale analysis of 25 recent LRMs, we reveal systematic failures to follow user constraints, particularly on harder problems and multi-constraint queries.
• We empirically demonstrate and dissect the *intelligence–obedience trade-off*, showing how reasoning-oriented training and longer CoTs simultaneously enhance problem-solving yet erode controllability.

## 2 RELATED WORK

### 2.1 LARGE REASONING MODELS (LRMS)

Recent advances in enhancing the reasoning ability of language models and reimplementing large reasoning models generally fall into two paradigms. The first paradigm constructs high-quality long CoT by distilling from more capable LRMs or combining primitive reasoning actions (Muennighoff et al., 2025; DeepSeek-AI, 2025). For example, s1 (Muennighoff et al., 2025) and LIMO (Ye et al., 2025) show that even a small amount of CoT data could significantly promote the reasoning ability. On the other hand, cold-RL on base language models directly attracts more and more attention in the subfield because of the success of deepseek-R1-zero and the previous findings that the model tends to memorize training data during the SFT process (Chu et al., 2025). In contrast with SFT, cold-RL does not rely on long CoT data and provides supervision signals by rewards on the final outcome (DeepSeek-AI, 2025) or the reasoning process (Liu et al., 2025). To simplify and accelerate the RL process, various techniques have been proposed, such as dynamic sampling (Yu et al., 2025), process-reward (Cui et al., 2025), off-policy guidance (Yan et al., 2025a), and CoT preference optimization (Yang et al., 2025). Recently, a concurrent work (Li et al., 2025b) also evaluates the instruction-following ability of LRMs. However, they evaluate on general-purpose benchmarks such as IFEval (Zhou et al., 2023) and ComplexBench (Wen et al., 2024), whereas we focus on LRMs whose training is predominantly math-oriented. To factor out confounding effects like domain mismatch, we design a dedicated testbed specifically for mathematical reasoning.

### 2.2 INSTRUCTION-FOLLOWING IN LLMS

As a crucial factor determining the practicality of a language model for real-world scenarios, the instruction-following ability is a core metric for language model evaluation, with numerous protocols and benchmarks being developed (Dubois et al., 2023; Chiang et al., 2023). Earlier benchmarks primarily focused on the completeness of user queries and depended on proprietary language models (Dubois et al., 2023; Chiang et al., 2023) to measure its win-rate over the baseline method, which is an oversimplification of real user queries. For a more comprehensive evaluation, sophisticated

benchmarks have been developed to test the ability of a language model in following format constraints (Zhou et al., 2023; Xia et al., 2024; Tang et al., 2024), multi-turn instruction (He et al., 2024c; Li et al., 2025a; Han, 2025; Sirdeshmukh et al., 2025), long-context instruction (Wu et al., 2024), multi-lingual instruction (He et al., 2024c; Li et al., 2025c), compositional instruction (Zhang et al., 2025; Hayati et al., 2025; Han, 2025) and refutation instructions (Yan et al., 2024; 2025b). More details about existing benchmarks is deferred to Appendix G. Most instruction-following benchmarks concentrate on the general domain and relatively straightforward queries. The domain shift and the lack of long CoT become a deterrent for using the benchmark on LRMs.

## 3 MATHIF

**Evaluating Instruction-following with General-purpose Benchmarks.** We begin by comparing several instruction-tuned LLMs with their reasoning-oriented counterparts. As shown in Figure 1, the same base model exhibits a clear performance drop in both IFEval (Zhou et al., 2023) and FollowBench (Jiang et al., 2023) after transitioning from the Instruct model to the reasoning model [1]. This observation suggests that reasoning-oriented training, while beneficial for problem-solving, may compromise instruction-following ability. However, since both benchmarks are designed for general-purpose tasks rather than mathematical reasoning, where LRMs are specifically

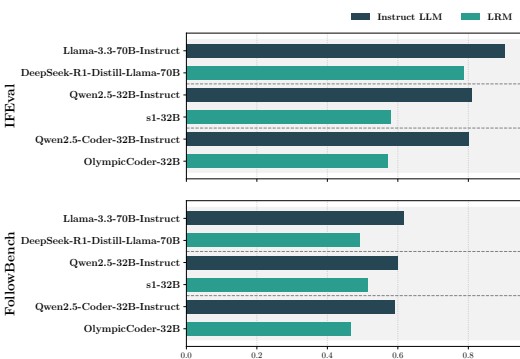

Figure 1: Performance of Instruct LLMs and LRMs on IFEval (Zhou et al., 2023) and Follow-Bench (Jiang et al., 2023).

optimized, it remains difficult to isolate instruction-following performance from confounding factors such as domain mismatch. To this end, we design **MathIF**, a dedicated testbed for evaluating the instruction-following ability of LRMs.

**Design Principles.** Our design follows several key principles tailored to mathematical reasoning: (1) evaluation is conducted entirely *within the math domain*, reducing confounding factors such as domain mismatch and allowing a sharper focus on the tension between reasoning and obedience; (2) all constraints are *objectively evaluable*, implemented as Python-verifiable rules to ensure deterministic and reproducible measurement; (3) the constraints are designed to *minimize interference with reasoning and answer extraction*: they apply only to the final answer segment (after the "</think>" tag) and largely involve lexical or formatting requirements, without altering how the reasoning process unfolds; (4) many constraints reflect *practical applicability*, such as token-length limits for latency control, bullet points and affixes for structured reporting, and language constraints for multilingual tutoring scenarios; (5) constraints are applied across problems of varying difficulty, from GSM8K to Olympiad and AIME, to enable a *difficulty-aware analysis* of the trade-off between reasoning accuracy and instruction adherence.

Building on these principles, we implement four categories of Python-verifiable constraints and compose instructions by combining two or three constraints at a time. These are embedded into math problems drawn from diverse sources spanning multiple difficulty levels. To support detailed evaluation, we further introduce two complementary metrics that enable fine-grained analysis of instruction-following performance in LRMs.

**Constraint Type.** Inspired by previous work (Zhou et al., 2023; Wen et al., 2024), we incorporate 15 constraints spanning four categories in our benchmark: (1) **Length constraints**, which limit response length to avoid excessive latency or token overhead at inference time, which is a common concern in deployment scenarios; (2) **Lexical constraints**, which require outputs in a specified language or mandate inclusion of key words/phrases, reflecting multilingual tutoring settings and keyword-driven educational tasks; (3) **Format constraints**, arguably the most frequent requests from real users, covering structured outputs such as a fixed number of sections, bullet points, punctuation usage, or

---

[1] DeepSeek-R1-Distill-Llama-70B, s1-32B and OlympicCoder-32B are trained from Llama-3.3-70B-Instruct, Qwen2.5-32B-Instruct and Qwen2.5-Coder-32B-Instruct, respectively.

Table 1: Dataset statistics grouped by source and by constraint.

| | Group by source | | | | | Group by constraint | | | Total |
|---|---|---|---|---|---|---|---|---|---|
| | **GSM8K** | **MATH500** | **Minerva** | **Olympiad** | **AIME** | **Single** | **Double** | **Triple** | |
| **# samples** | 90 | 90 | 90 | 90 | 60 | 140 | 140 | 140 | 420 |
| **Avg. Len** | 86.73 | 57.24 | 88.09 | 80.42 | 87.25 | 64.89 | 83.84 | 89.54 | 79.43 |

case sensitivity, all of which are critical for downstream reporting or documentation pipelines; and (4) **Affix constraints**, which demand specific prefixes, suffixes, or both, ensuring models can reliably wrap responses with required tokens or phrases, which is useful in templated applications like chatbots and automated grading systems. To ensure objective and reliable evaluation, all constraint compliance in MathIF is verified deterministically using Python scripts. A more detailed categorization for the type and subtype of constraints is listed in Appendix F, together with an illustrating example and the entire list of constraints.

**Compositional Constraint.** Queries with only a single constraint can hardly reflect the complex scenarios encountered by a downstream application of LRM, as the real user queries to LRMs typically contain more than one restrictive condition (Wen et al., 2024). Therefore, we construct compositional constraints by combining two or three individual constraints. Specifically, given the set of individual constraints denoted as $\mathcal{C}$, we enumerate all the elements in the Cartesian product $\mathcal{C}^2 = \{(c_1, c_2) \mid c_1, c_2 \in \mathcal{C}\}$ and $\mathcal{C}^3 = \{(c_1, c_2, c_3) \mid c_1, c_2, c_3 \in \mathcal{C}\}$, from which we randomly sample several combinations after manually filtering out the ones in which the constraints are incompatible with each other and fall into the same subtype of constraint. Through this procedure, we harvest 30 dual-constraints and 15 triple-constraints. The detailed list of dual-constraints and triple-constraints is presented in Table 12.

**Math Problem Collection.** With the constructed individual constraints and compositional constants, the next step is to incorporate these constraints into math problems to constitute a query. To systematically assess instruction-following across problem difficulty, MathIF contains math problems of varying levels of difficulty, ranging from math word problems in primary school and math problems in high school to the latest math problems in world-level competition. Specifically, we randomly sample 90 problems from GSM8K (Cobbe et al., 2021), MATH-500 (Hendrycks et al., 2021), Minerva (De et al., 2013), Olympiad (He et al., 2024a) respectively. For AIME2024&2025 (Veeraboina, 2023), we use all the 60 problems it contained. For each data source, we apply a single constraint, dual constraints, and triple constraints, resulting in three subsets of equivalent size. For a sanity check, we manually review the curated samples and double-check whether the added constraints are contradictory to the math problem itself. The statistics for the established dataset are shown in Table 1.

**Evaluation Metric** To systematically measure whether one or more constraints in the query are satisfied by the LRM while solving the math problems, we follow previous works (Zhou et al., 2023; Jiang et al., 2023) and use two metrics of different granularity. Specifically, we employ **hard accuracy (HAcc)** and **soft accuracy (SAcc)** to measure whether the model response follows the constraints at the query level and constraint level, respectively. Formally, suppose a query has $n$ constraints $\mathcal{C}_1, \mathcal{C}_2, \mathcal{C}_3, \ldots, \mathcal{C}_n$ and we use $\mathbb{I}(\mathcal{C}_i)$ to denote whether the $i$-th constraint is satisfied or not, with $\mathbb{I}(\mathcal{C}_i) = 1$ for satisfied constraint and $\mathbb{I}(\mathcal{C}_i) = 0$ for unsatisfied constraint. The hard accuracy (HAcc) and soft accuracy (SAcc) for a query is defined as:

$$\text{HAcc} = \prod_{i=1}^{n} \mathbb{I}(\mathcal{C}_i), \quad \text{SAcc} = \frac{1}{n} \sum_{i=1}^{n} \mathbb{I}(\mathcal{C}_i) \tag{1}$$

Notably, for queries with only a single constraint, the two metrics are identical in number. The overall hard accuracy and soft accuracy on the benchmark are averaged among all the queries in the dataset. Apart from instruction-following ability, we also measure the correctness of the math problem solution on our proposed MathIF, defined as whether the final answer exactly matches the ground-truth, regardless of constraint satisfaction. By default, correctness refers to performance with constraints in the prompts unless specified (e.g., Table 2).

Table 2: Experimental results of LRMs on MathIF. We report hard accuracy (HAcc) and soft accuracy (SAcc) for instruction-following, alongside math-solving correctness *with* and *without* constraints (w/o const. / w/ const.). The last column shows the relative change in correctness when constraints are included. Models are sorted in descending order of instruction-following performance. † indicates models trained by supervised fine-tuning only (no reasoning-oriented RL). **Bold** and underlined values denote the *top*-2 and *bottom*-2 entries for open-sourced models in each column, respectively.

| Model | Instruction Following | | Correctness | | |
| | HAcc | SAcc | w/o const. | w/ const. | Diff.(%) |
|---|---|---|---|---|---|
| *Models with no more than 4B parameters* | | | | | |
| Qwen3-4B | **44.05** | **61.43** | **68.10** | **58.57** | **-13.99** |
| Qwen3-1.7B | **30.24** | 50.24 | **62.38** | **51.19** | -17.94 |
| Qwen3-0.6B | 27.86 | **50.44** | 40.95 | 32.14 | -21.51 |
| L1-Qwen-1.5B-Exact | 19.76 | 39.60 | 53.81 | 42.86 | -20.35 |
| L1-Qwen-1.5B-Max | 19.76 | 39.40 | 55.48 | 45.71 | -17.61 |
| DeepSeek-R1-Distill-Qwen-1.5B† | 17.14 | 36.62 | 52.86 | 31.67 | -40.09 |
| DeepScaler-1.5B-Preview | 14.52 | 34.52 | 58.10 | 36.19 | -37.71 |
| Qwen2.5-1.5B-SimpleRL-Zoo | 9.05 | 24.33 | 27.14 | 22.38 | -17.54 |
| Qwen2.5-Math-1.5B-Instruct | 7.62 | 21.39 | 44.05 | 44.29 | **+0.54** |
| *Models with approximately 7B–14B parameters* | | | | | |
| Qwen3-14B | **50.71** | **67.06** | 71.43 | **64.29** | -10.00 |
| DeepSeek-R1-Distill-Qwen-14B† | **39.28** | **60.55** | 67.14 | 50.95 | -24.11 |
| Qwen3-8B | 37.86 | 57.34 | **69.52** | **66.43** | **-4.44** |
| DeepSeek-R1-Distill-Qwen-7B† | 26.43 | 44.96 | 65.24 | 48.57 | -25.55 |
| DeepSeek-R1-Distill-Llama-8B† | 22.14 | 44.04 | 59.76 | 36.43 | -39.04 |
| Open-Reasoner-Zero-7B | 13.57 | 32.26 | 52.86 | 51.90 | **-1.82** |
| Qwen2.5-Math-7B-Instruct | 9.05 | 25.60 | 46.90 | 37.14 | -20.81 |
| *Models with 32B or more parameters* | | | | | |
| Qwen3-32B | **43.81** | **62.82** | **72.62** | **70.00** | -3.61 |
| DeepSeek-R1-Distill-Qwen-32B† | **42.62** | 60.91 | **71.43** | 57.62 | -19.33 |
| DeepSeek-R1-Distill-Llama-70B† | 41.43 | **61.07** | 71.19 | 54.05 | -24.08 |
| QwQ-32B | 40.24 | 59.99 | 70.95 | **68.81** | **-3.02** |
| OlympicCoder-32B | 35.95 | 57.97 | 59.29 | 54.52 | -8.05 |
| s1-32B† | 20.95 | 41.78 | 62.86 | 60.95 | -3.04 |
| Open-Reasoner-Zero-32B | 15.47 | 35.52 | 65.48 | 67.62 | **+3.27** |
| *Close-sourced Commercial Models* | | | | | |
| o3-mini | 78.81 | 87.30 | 65.24 | 65.95 | +0.71 |
| Gemini-2.5-pro-preview | 70.71 | 81.79 | 66.19 | 68.33 | +2.14 |

## 4 EXPERIMENT

To benchmark the instruction-following ability of LRMs, we evaluate a diverse set of models across three parameter scales. We follow previous work (Zeng et al., 2025) and adopt a commonly used generation parameters using nucleus sampling ($T$=1.0, $p$=0.95) with a maximum generation length of 16,384 tokens, powered by the vLLM (Kwon et al., 2023) engine for efficient inference.

• **Small-scale models ($\leq$ 4B parameters):** Qwen3-0.6B (Team, 2025b), Qwen2.5-1.5B-SimpleRL-Zoo (Zeng et al., 2025), Qwen2.5-Math-1.5B-Instruct (Yang et al., 2024), DeepSeek-R1-Distill-Qwen-1.5B (DeepSeek-AI, 2025), DeepScaler-1.5B-Preview (Luo et al., 2025), L1-Qwen-1.5B-Max (Aggarwal & Welleck, 2025), L1-Qwen-1.5B-Exact (Aggarwal & Welleck, 2025), Qwen3-1.7B (Team, 2025b), Qwen3-4B (Team, 2025b).

• **Medium-scale models (7B∼14B parameters):** Qwen2.5-Math-7B-Instruct (Yang et al., 2024), DeepSeek-R1-Distill-Qwen-7B (DeepSeek-AI, 2025), Open-Reasoner-Zero-7B (Hu et al., 2025a), DeepSeek-R1-Distill-Llama-8B (DeepSeek-AI, 2025), Qwen3-8B (Team, 2025b), DeepSeek-R1-Distill-Qwen-14B (DeepSeek-AI, 2025), Qwen3-14B (Team, 2025b).

- **Large-scale models (≥ 32B parameters):** s1-32B (Muennighoff et al., 2025), OlympicCoder-32B (Face, 2025), DeepSeek-R1-Distill-Qwen-32B (DeepSeek-AI, 2025), QwQ-32B (Team, 2025c), Open-Reasoner-Zero-32B (Hu et al., 2025b), Qwen3-32B (Team, 2025b), DeepSeek-R1-Distill-Llama-70B (DeepSeek-AI, 2025).
- **Close-sourced Commercial Models:** o3-mini (OpenAI) and Gemini-2.5-pro-preview (Team, 2025a).

## 4.1 Experimental Results

The experimental results, as summarized in Table 2, reveal several key factors that influence the instruction-following performance of LRMs:

**All LRMs fail to obey most user instructions.** All LRMs evaluated on MathIF exhibit poor instruction-following performance. Even the best-performing model, Qwen3-14B, achieves only 50.71% hard accuracy, barely surpassing the halfway mark. The majority of models, including large-scale variants such as DeepSeek-R1-Distill-Llama-70B and Open-Reasoner-Zero-32B, fail to meet even minimal expectations for faithfully executing user-specified constraints.

**Model scale alone does not determine instruction-following performance.** While larger models often perform better within the same series (e.g., Qwen2.5-Math and Open-Reasoner-Zero), scaling up does not guarantee improvement across different architectures. For instance, DeepSeek-R1-Distill-Llama-70B underperforms Qwen3-4B despite being more than $15\times$ larger. Notably, Qwen3-8B and Qwen3-32B deviate from the within-series scaling trend, highlighting that instruction-following ability depends on both model size and design.

**There exists a trade-off between instruction-following and mathematical reasoning.** As shown in the "Diff" column of Table 2, most models experience a drop in problem-solving correctness when additional constraints are introduced, with margins ranging from 0.96 to 23.33. This suggests that stronger adherence to external constraints may compromise core mathematical reasoning. The only exceptions are Qwen2.5-Math-1.5B-Instruct and Open-Reasoner-Zero-32B, which maintain or slightly improve their performance under constrained conditions.

We view the performance of o3-mini and Gemini-2.5 as an approximate upper bound under current proprietary pipelines. Their strong results further validate the importance of instruction-following evaluation, but they do not diminish our finding: across open LRMs, scaling reasoning consistently degrades controllability unless specialized alignment interventions are applied.

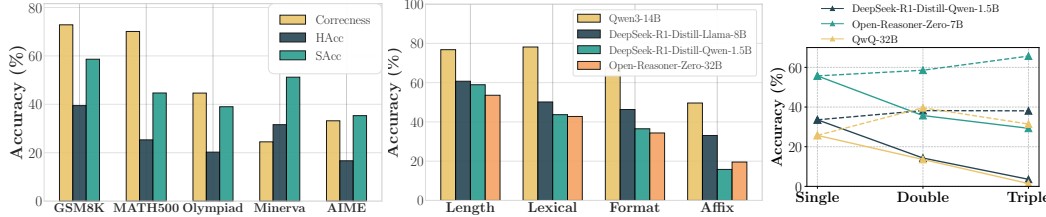

Figure 2: **Left**: The accuracy on each math subset averaged over models; **Middle**: HAcc on each constraint subset averaged over models; **Right**: SAcc (solid line) and SAcc (dashed line) on the single/double/triple-constraint subsets.

**Dissecting Instruction-Following Performance.** We first scrutinize the model performance on each subset and visualize the average accuracy of tested LRMs in Figure 2 (left). We can observe a performance difference among different subsets and whether an LRM follows the constraints is correlated with the difficulty level of the math problem with easier math problems being more likely to be followed. Turning to Figure 2 (middle), we observe that **length constraints** are easiest to satisfy, while lexical and format constraints demand finer token-level control and thus reduce accuracy. Affix constraints prove most difficult, highlighting that constraint type itself—beyond problem difficulty—strongly shapes instruction-following performance. Next, we investigate the impact of the constraint number and plot the instruction-following accuracy of three LRMs in Figure 2 (right). We can observe an obvious deterioration in hard accuracy when increasing the number of constraints but the soft accuracy remains unchanged or slightly fluctuated. It seems that the model's ability to follow every individual constraint can be enhanced by the existence of more constraints. Please refer to Appendix D for more details.

# 5 WHEN SCALING REASONING MEETS LOSING CONTROL

As discussed in Section 4.1, there may exist a trade-off between the instruction-following ability and the mathematical reasoning capability of LRMs. In this section, we further investigate this trade-off through a fine-grained error analysis (Section 5.1), examine the effects of different reasoning-oriented training paradigms (Section 5.2), and explore how CoT length impacts reasoning and instruction-following by applying both inference-time and training-aware interventions (Section 5.3).

## 5.1 THE INTELLIGENCE–OBEDIENCE TRADE-OFF

**Dilemma between Reasoning and Instruction Following.** We begin by analyzing the relationship between reasoning and instruction-following through an error-based categorization. Each sample is grouped into one of four categories based on two criteria: (1) whether the math problem was solved correctly, and (2) whether all user-specified constraints were satisfied. The proportions of these four categories are shown in Figure 3. We observe that LRMs often struggle to fulfill both objectives simultaneously, as evidenced by the particularly low

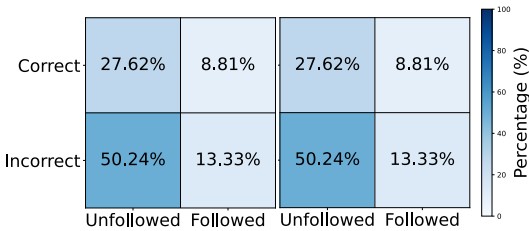

Figure 3: Error analysis of DeepSeek-R1-Distill-Llama-8B (left) and Qwen3-32B (right) on each subset of MathIF.

proportion of (Correct, Followed) cases. Interestingly, the proportion of (Correct, Followed) is even smaller than that of either (Correct, Unfollowed) or (Incorrect, Followed), suggesting that LRMs frequently sacrifice one objective to achieve the other, consistent with the trend in Table 2.

Table 2 (last column) shows a noticeable degradation in math problem correctness when constraints are introduced. Figure 4 further breaks down this effect by dataset. Surprisingly, we find that the drop rate on GSM8K (the easiest subset) is even higher than that on AIME (the hardest), a significantly more challenging benchmark. This suggests that the impact of constraints on reasoning performance is not necessarily correlated with problem difficulty. In conclusion, the trade-off between instruction-following and reasoning appears to be a general phenomenon across dif-

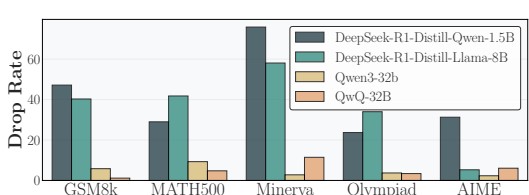

Figure 4: Relative correctness drop of four LRMs across five subsets.

ficulty levels. Notably, LRMs fine-tuned on long CoT traces (e.g., DeepSeek-R1 variants) tend to exhibit more severe performance degradation than RL-trained models like Qwen3-32B and QwQ-32B, possibly due to the inherent limitations of SFT (Chu et al., 2025).

**Longer CoTs Impair Instruction Following.** We further analyze the impact of CoT length on instruction-following performance. Specifically, for each LRM, we divide its benchmark outputs into six bins based on the number of tokens between the `<think>` and `</think>` delimiters. The resulting trends are shown in Figure 5. Across all three models, i.e, DeepSeek-R1-Distill-Llama-8B, Qwen3-0.6B, and Qwen3-32B, we observe a consistent decline in both hard accuracy and soft accuracy as CoT length increases, suggesting a negative correlation between generation length and instruction compliance. One possible explanation is that longer CoTs, while beneficial for reasoning, increase the distance between the user-specified constraint and the final answer. This may dilute the model's attention to the constraint, making accurate instruction-following more difficult (see Section 5.3).

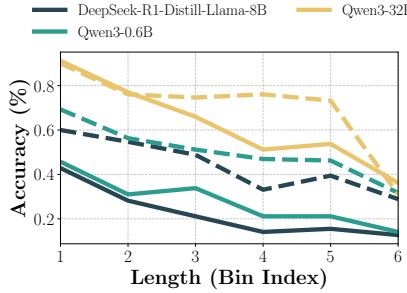

Figure 5: HAcc (solid line) and SAcc (dashed line) across six CoT length bins; higher indices correspond to longer CoT generations.

## 5.2 How Does Reasoning-Oriented Training Affect Instruction-Following?

Motivated by the patterns observed in Figure 4, we further investigate how different reasoning-oriented training paradigms affect a model's instruction-following behavior. Specifically, we examine three representative strategies: (1) **SFT-only**, (2) **SFT followed by RL** (SFT+RL), and (3) **cold-start RL** (i.e., zero-RL), which bypasses SFT entirely.

**Training Setup.** We base our experiments on the DeepScaler dataset (Luo et al., 2025), which contains approximately 40k math reasoning samples. All training is conducted using 16 NVIDIA H100 GPUs. For SFT-only and SFT+RL settings, we first distill long CoT reasoning traces from QwQ-32B (Team, 2025c), filtering out samples where QwQ-32B fails to generate a correct answer or the CoT exceeds 8192 tokens. This results in 18k high-quality examples. We use models from the Qwen-2.5 and Qwen-2.5-Math series as our base. Since some models are limited to 4096 position embeddings, we extend the RoPE (Su et al., 2024) scaling factor $\theta$ from 10,000 to 20,000 to accommodate longer sequences, following prior work (Yan et al., 2025a). For reinforcement learning, we adopt the GRPO (Shao et al., 2024) framework and use verifiable outcome-based rewards. In addition to standard correctness rewards, we design a format-aware reward variant (**w/ format reward**) that grants 0.1 if the model includes special reasoning tokens (e.g., `<think>` and `</think>`) and 1.0 for a correct solution.

Table 3: Comparison of reasoning-oriented training strategies. Avg. Acc. denotes math reasoning performance (more details in Appendix E). Cells shaded in green and red indicate increased and decreased instruction-following performance, respectively, relative to the base model.

| Model | HAcc | SAcc | Corectness |
|---|---|---|---|
| **Qwen2.5-1.5B** | 10.00 | 27.26 | 1.21 |
| +SFT | 7.86 | 22.70 | 4.20 |
| +SFT+RL | 7.86 | 20.44 | 12.54 |
| +cold-RL | 9.52 | 23.97 | 14.58 |
| w/ format reward | 10.95 | 28.49 | 11.17 |
| **Qwen-2.5-7B** | 15.95 | 33.13 | 13.59 |
| +SFT | 7.86 | 21.03 | 23.10 |
| +SFT+RL | 7.62 | 21.07 | 32.82 |
| +cold-RL | 10.48 | 27.26 | 28.39 |
| w/ format reward | 14.52 | 32.50 | 24.80 |
| **Qwen2.5-Math-1.5B** | 9.28 | 23.33 | 18.91 |
| +SFT | 7.86 | 21.03 | 14.39 |
| +SFT+RL | 7.14 | 20.56 | 24.71 |
| +cold-RL | 8.33 | 21.31 | 24.88 |
| w/ format reward | 7.62 | 20.08 | 23.95 |
| **Qwen2.5-Math-7B** | 9.76 | 23.53 | 20.68 |
| +SFT | 8.09 | 22.06 | 29.11 |
| +SFT+RL | 8.57 | 21.03 | 40.65 |
| +cold-RL | 7.85 | 22.62 | 32.61 |
| w/ format reward | 7.86 | 21.79 | 32.66 |

**The Double-Edged Sword of Reasoning-Oriented Training.** Table 3 presents the results for different training pathways, where Corectness denotes overall math reasoning performance (details in Appendix E). While both SFT and RL reliably boost reasoning accuracy, neither improves instruction-following. Instead, we observe a consistent—and in some cases substantial—decline in HAcc and SAcc, *with trained models even performing worse than their base model counterparts*. For example, Qwen2.5-1.5B and Qwen2.5-7B both lose more than 10 points in SAcc after SFT or RL despite clear reasoning gains. The format-aware reward yields slight improvements for Qwen-2.5-1.5B, 7B but has negligible effect on the Math series. These results show that reasoning-oriented post-training does not merely overlook obedience but can actively erode it, revealing a central *dilemma* in current training paradigms: *sharpening intelligence often comes at the expense of control*.

## 5.3 How does the CoT Length Affect Instruction Following?

**The More Thinking, the Less Following.** To investigate how CoT length influences instruction adherence, we artificially increase the CoT length using budget forcing (Muennighoff et al., 2025), which appends the token "Wait" each time the model attempts to terminate the reasoning process. This encourages the model to continue generating longer CoTs. The experiment is performed on DeepSeek-R1-Distill-Qwen-1.5B, and Figure 6 shows the instruction-following performance as the number of budget-forcing steps $N$ increases from 2 to 8. As CoT length increases, SAcc steadily declines, suggesting that excessively long CoTs may impair the model's ability to

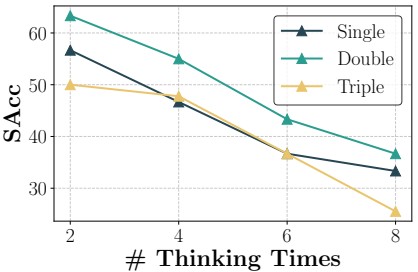

Figure 6: The trend of SAcc on GSM8K subset as the number of "Wait" rethinking increases from 2 to 8.

follow instructions. This degradation likely stems from the increasing distance between the instruction and the final output, which may dilute the model's attention to user constraints (Li et al., 2025b).

**Controlling CoT Length During RL Training.**   Beyond inference-time manipulation, we investigate whether controlling the length of CoT during reinforcement learning has a similar impact on instruction-following. Specifically, we continue RL training on DeepSeek-R1-Distill-Qwen-1.5B using the DeepScaler dataset (Luo et al., 2025), varying the maximum response length during rollouts.

In this setup, overlong responses are truncated and receive no outcome reward, encouraging the model to response within allowed length. We adopt a pure outcome-based reward function and conduct RL training for three epochs, varying the maximum rollout length from 1k to 8k tokens. The results, shown in Table 4, reveal a clear trend: as the maximum rollout length increases, math reasoning performance (averaged across AIME2024, AIME2025, AMC2023, Minerva, and Olympiad, more details in Appendix E) improves, while both hard accuracy and soft accuracy consistently decline. This ob-

Table 4: Impact of the maximum response length during RL. Cells shaded in red denote lower performance relative to the base (*Original*), with intensity proportional to the drop magnitude.

| Model | HAcc | SAcc | Avg. Acc. |
|---|---|---|---|
| Original | 17.14 | 36.62 | 36.13 |
| +cold-RL (1k) | 19.05 | 39.88 | 28.73 |
| +cold-RL (2k) | 16.43 | 36.75 | 36.32 |
| +cold-RL (4k) | 16.91 | 35.87 | 40.03 |
| +cold-RL (8k) | 14.29 | 34.13 | 39.82 |

servation further reinforces our conclusion: *reasoning-oriented training that **favors longer CoTs** can inadvertently **harm instruction-following** fidelity, highlighting a persistent trade-off between reasoning strength and obedience to user constraints.*

**Bringing Instructions Closer Improves Obedience at the Cost of Intelligence.**   One possible explanation for the negative impact of lengthy CoTs on instruction-following is that extended reasoning increases the distance between the user query and the final answer, making it more likely for the model to overlook the original constraint. To preliminarily verify this hypothesis, we propose a simple yet effective remedy: repeating the con-

Table 5: Effect of `+repeat` on model performance. Cells shaded in red/green denote lower/higher performance relative to vanilla generation.

| Model | HAcc | SAcc | Correctness |
|---|---|---|---|
| DeepSeek-R1-Distill-Qwen-1.5B | 17.14 | 36.62 | 31.67 |
| +repeat | 21.66 | 42.58 | 22.38 |
| Open-Reasoner-Zero-7B | 13.57 | 32.26 | 51.90 |
| +repeat | 14.53 | 33.14 | 30.00 |
| Qwen3-32B | 43.81 | 62.82 | 70.00 |
| +repeat | 59.29 | 68.34 | 63.81 |

straint at the end of the CoT. Concretely, we manually append the token "Wait" to prolong the CoT and then **reintroduce** the original constraint immediately afterward. As a result, the constraint appears twice in the input, i.e., once before the CoT begins and once again at the end, thereby shortening its contextual distance from the final answer. Experimental results on DeepSeek-R1-Distill-Qwen-1.5B, Open-Reasoner-Zero-7B, and Qwen3-32B are shown in Table 5. This straightforward intervention leads to clear improvements in instruction-following (SAcc and HAcc), albeit at a modest cost to problem-solving accuracy. These findings further confirm the inherent trade-off between reasoning depth and obedience during inference: *enhancing one often comes at the expense of the other.*

## 6   CONCLUSION

Our study reveals a persistent and underexplored trade-off between reasoning strength and instruction-following fidelity. Through MathIF, a benchmark tailored for evaluating instruction adherence in math reasoning tasks, we show that reasoning scaling does not guarantee control. Empirical results reveal that longer chains of thought and reasoning-oriented training methods (e.g., SFT and RL) often impair a model's ability to comply with user-specified constraints. These findings highlight a core tension in the development of LRMs: as models become more intelligent, they often become less controllable. This dilemma is central to the alignment problem in reasoning-centric systems. Addressing it requires rethinking current training paradigms to build models that can reason effectively without drifting from user intent. We hope that our benchmark and findings serve as a foundation for future research that bridges the growing gap between intelligence and obedience in large reasoning models.

## ETHICS STATEMENT

Our proposed MathIF evaluates the instruction-following ability of publicly released LRMs, adhering strictly to the ICLR Code of Ethics. The math problems used for our benchmark are collected from free public datasets, and the construction of our benchmark does not involve recruiting crowdsource workers or human annotators. Our benchmark should only be used for research, not for any malicious purpose.

## REPRODUCIBILITY STATEMENT

To ensure the reproducibility of our results, we introduce the experimental setup in Section 4 and elaborate on the hyper-parameter setting and poison data construction in Appendix C. The construction process of our benchmark is elaborated on Section 3 and a full list of constraint used in our benchmark could be found in Appendix F. The anonymous code for benchmark evaluation could be downloaded from `https://anonymous.4open.science/r/MathIF-v1-0859`.

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

## A  OVERVIEW OF THE APPENDIX

This Appendix is organized as follows:

- Section B and Section H discussed the limitation and the use of LLM in our study, respectively.
- Section C elaborate on the hyper-parameters used for our reasoning-oriented training in Section 5.2.
- Section D provides more detailed results on our benchmark to facilitate analysis on the difficulty of math problems and the number of constraints.
- Section E contains detailed reasoning performance for LRMs trained in Section 5.
- Section F lists the constraints used in our proposed MathIF benchmark and provides a fine-grained analysis.
- Section G provides a more comprehensive review of existing instruction-following benchmarks.

## B  LIMITATIONS

The limitations of this study can be summarized as below:

- In this study, we evaluate 23 recently released LRMs in text modality, and we plan to leave the benchmarking of large vision reasoning models for future work.
- When investigating how reasoning-oriented training affects instruction-following, we mainly use GRPO (Shao et al., 2024) for RL training because of its simplicity, stability, and widespread practical adoption. Experimenting with other RL training algorithms is left for future work.

## C  HYPER-PARAMETER SETTING

Our experiments on different reasoning-oriented training strategies in Section 5.2 are conducted on a cloud Linux server with Ubuntu 16.04 operating system. The codes are written in Python 3.10 with the huggingface libraries[2]. We run our experiments on 16 Nvidia H100 with 80GiB GPU memory. The detailed hyper-parameter settings for supervised fine-tuning and reinforcement learning are shown in Table 6, which mostly follow the default setting in VeRL framework [3].

Table 6: The value of the hyper-parameters in our reasoning-oriented training experiment (Section 5.2) for SFT (left) and RL (right).

| Hyper-parameter | Value |
|---|---|
| batch_size | 256 |
| micro_batch_size | 1 |
| max_length | 8192 |
| rope_theta | 20000 |
| lr | 1e-6 |
| betas | (0.9, 0.95) |
| weight_decay | 0.01 |
| warmup_ratio | 0.1 |
| schedule | cosine |
| clip_grad | 1 |
| epoch | 3 |
| truncation | right |
| sliding_window | none |

| Hyper-parameter | Value |
|---|---|
| max_prompt_length | 1024 |
| max_response_length | 3072 |
| lr | 1e-6 |
| batch_size | 128 |
| mini_batch_size | 64 |
| grad_clip | 1 |
| clip_ratio | 0.2 |
| entropy_coeff | 0.001 |
| kl_loss_coef | 0.001 |
| rl_epoch | 1 |
| warmup_ratio | 0 |
| schedule | constant |
| rollout_n | 8 |
| rollout_temperature | 1 |

## D  MORE BENCHMARK RESULTS

In Section 4.3, we visualize the model performance grouped by the source of math problems and the number of constraints. In this section, we supplement with more detailed benchmark results

---

[2] https://github.com/huggingface/transformers
[3] https://github.com/volcengine/verl

for LRMs involved in our experiments. The fine-grained instruction-following performance across different source of math problems are presented in Table 7, while the hard accuracy (HAcc) and soft accuracy (SAcc) for different number of constraints are shown in Table 8 and Table 9, respectively.

Table 7: Experimental results of LRMs on MathIF. We report hard accuracy (HAcc) and soft accuracy (SAcc) for instruction-following. † indicates models trained by supervised fine-tuning only (no reasoning-oriented RL).

| Model | single | double | | triple | |
| | Acc | HAcc | SAcc | HAcc | SAcc |
| --- | --- | --- | --- | --- | --- |
| Models with no more than 4B parameters | | | | | |
| Qwen3-4B | 53.57 | 38.57 | 57.86 | 40.00 | 72.86 |
| Qwen3-1.7B | 42.14 | 22.86 | 46.43 | 25.71 | 62.14 |
| Qwen3-0.6B | 48.57 | 22.86 | 48.93 | 12.14 | 53.81 |
| L1-Qwen-1.5B-Exact | 33.57 | 18.57 | 43.57 | 7.14 | 41.66 |
| L1-Qwen-1.5B-Max | 37.14 | 16.43 | 43.93 | 5.71 | 37.14 |
| DeepSeek-R1-Distill-Qwen-1.5B† | 33.57 | 14.29 | 38.21 | 3.57 | 38.09 |
| DeepScaler-1.5B-Preview | 30.71 | 10.00 | 35.00 | 2.86 | 37.85 |
| Qwen2.5-1.5B-SimpleRL-Zoo | 21.43 | 2.86 | 21.07 | 2.86 | 30.48 |
| Qwen2.5-Math-1.5B-Instruct | 19.29 | 2.14 | 19.64 | 1.43 | 25.24 |
| Models with approximately 7B–14B parameters | | | | | |
| Qwen3-14B | 63.57 | 40.71 | 60.71 | 47.86 | 76.90 |
| DeepSeek-R1-Distill-Qwen-14B† | 57.14 | 35.71 | 62.86 | 25.00 | 61.66 |
| Qwen3-8B | 51.43 | 31.43 | 54.64 | 30.71 | 65.95 |
| DeepSeek-R1-Distill-Qwen-7B† | 39.29 | 27.14 | 50.36 | 12.86 | 45.23 |
| DeepSeek-R1-Distill-Llama-8B† | 34.29 | 22.14 | 47.14 | 10.00 | 50.7 |
| Open-Reasoner-Zero-7B | 25.71 | 13.57 | 39.64 | 1.43 | 31.42 |
| Qwen2.5-Math-7B-Instruct | 22.86 | 2.86 | 24.64 | 1.43 | 29.29 |
| Models with 32B or more parameters | | | | | |
| Qwen3-32B | 61.43 | 35.00 | 57.50 | 35.00 | 69.52 |
| DeepSeek-R1-Distill-Qwen-32B† | 57.14 | 37.14 | 60.36 | 33.57 | 65.23 |
| DeepSeek-R1-Distill-Llama-70B† | 54.29 | 39.29 | 61.07 | 30.71 | 67.85 |
| QwQ-32B | 55.71 | 35.71 | 58.57 | 29.29 | 65.69 |
| OlympicCoder-32B† | 55.71 | 31.43 | 60.36 | 20.71 | 57.85 |
| s1-32B† | 37.14 | 13.57 | 38.93 | 12.14 | 49.27 |
| Open-Reasoner-Zero-32B | 30.71 | 13.57 | 41.79 | 2.14 | 34.05 |

## E  MORE RESULTS ON MATH BENCHMARKS

In Section 5.2, we vary the reasoning-oriented training strategy and report the averaged math reasoning performance among five benchmarks in Table 3. The five benchmarks used in our experiments are: AIME2024 [4], AIME2025 [5], AMC2023 [6], Minerva [7], and Olympiad [8]. For fine-grained analysis, we report more detailed results on five benchmarks in Table 11. Similarly, in Section 5.3, we control the CoT length during RL training and report the averaged math reasoning performance among the five benchmarks in Table 4 and detailed results on five benchmarks in Table 10.

## F  ANALYSIS ON CONSTRAINT TYPES

In this section we provide a detailed list of the 15 constraints used in our benchmark in Table **??** and Table 13, together with the instruction-following performance per constraint type in Table 14. From the table, we could observe that the performance on the length constraint and the lexical constraint is substantially better, while the performance on the Affix constraint is the worst among the four categories.

---

[4] https://huggingface.co/datasets/HuggingFaceH4/aime_2024

[5] https://huggingface.co/datasets/opencompass/AIME2025

[6] https://huggingface.co/datasets/zwhe99/amc23

[7] https://huggingface.co/datasets/math-ai/minervamath

[8] https://huggingface.co/datasets/zwhe99/simplerl-OlympiadBench

Table 8: Experimental results of LRMs on MathIF. We report hard accuracy (HAcc) for instruction-following on five subsets of our MathIF. † indicates models trained by supervised fine-tuning only (no reasoning-oriented RL).

| Model | GSM8K | MATH500 | Minerva | Olympiad | AIME |
|---|---|---|---|---|---|
| Models with no more than 4B parameters | | | | | |
| Qwen3-4B | 66.67 | 40.00 | 53.33 | 31.11 | 21.67 |
| Qwen3-1.7B | 44.44 | 25.56 | 41.11 | 24.44 | 8.33 |
| Qwen3-0.6B | 36.67 | 25.56 | 34.44 | 24.44 | 13.33 |
| L1-Qwen-1.5B-Exact | 27.78 | 15.56 | 21.11 | 17.78 | 15.00 |
| L1-Qwen-1.5B-Max | 24.44 | 18.89 | 22.22 | 16.67 | 15.00 |
| DeepSeek-R1-Distill-Qwen-1.5B† | 32.22 | 12.22 | 15.56 | 12.22 | 11.67 |
| DeepScaler-1.5B-Preview | 26.67 | 10.00 | 15.56 | 7.78 | 11.67 |
| Qwen2.5-1.5B-SimplRL-Zoo | 11.11 | 10.00 | 11.11 | 4.44 | 8.33 |
| Qwen2.5-Math-1.5B-Instruct | 8.89 | 5.56 | 8.89 | 6.67 | 8.33 |
| Models with approximately 7B–14B parameters | | | | | |
| Qwen3-14B | 71.11 | 53.33 | 63.33 | 35.56 | 20.00 |
| DeepSeek-R1-Distill-Qwen-14B† | 55.56 | 35.56 | 44.44 | 31.11 | 25.00 |
| Qwen3-8B | 56.67 | 37.78 | 44.44 | 24.44 | 20.00 |
| DeepSeek-R1-Distill-Qwen-7B† | 46.67 | 22.22 | 31.11 | 14.44 | 13.33 |
| DeepSeek-R1-Distill-Llama-8B† | 41.11 | 18.89 | 20.00 | 13.33 | 15.00 |
| Open-Reasoner-Zero-7B | 13.33 | 14.44 | 11.11 | 13.33 | 16.67 |
| Qwen2.5-Math-7B-Instruct | 12.22 | 5.56 | 10.00 | 8.89 | 8.33 |
| Models with 32B or more parameters | | | | | |
| Qwen3-32B | 73.33 | 40.00 | 52.22 | 26.67 | 18.33 |
| DeepSeek-R1-Distill-Qwen-32B† | 57.78 | 38.89 | 52.22 | 32.22 | 26.67 |
| DeepSeek-R1-Distill-Llama-70B† | 55.56 | 42.22 | 53.33 | 28.89 | 20.00 |
| QwQ-32B | 60.00 | 38.89 | 45.56 | 32.22 | 16.67 |
| OlympicCoder-32B† | 36.67 | 36.67 | 37.78 | 31.11 | 38.33 |
| s1-32B† | 33.33 | 20.00 | 22.22 | 13.33 | 13.33 |
| Open-Reasoner-Zero-32B | 15.56 | 14.44 | 15.56 | 14.44 | 18.33 |

Table 9: Experimental results of LRMs on MathIF. We report soft accuracy (SAcc) for instruction-following on five subsets of our MathIF. † indicates models trained by supervised fine-tuning only (no reasoning-oriented RL).

| Model | GSM8K | MATH500 | Minerva | Olympiad | AIME |
|---|---|---|---|---|---|
| Models with no more than 4B parameters | | | | | |
| Qwen3-4B | 80.19 | 57.41 | 70.37 | 50.19 | 42.78 |
| Qwen3-1.7B | 65.74 | 44.81 | 61.85 | 45.19 | 25.28 |
| Qwen3-0.6B | 61.30 | 47.04 | 59.07 | 45.37 | 33.89 |
| L1-Qwen-1.5B-Exact | 50.56 | 37.59 | 39.62 | 33.7 | 35.00 |
| L1-Qwen-1.5B-Max | 45.37 | 40.56 | 42.78 | 34.44 | 31.10 |
| DeepSeek-R1-Distill-Qwen-1.5B† | 54.26 | 32.59 | 37.03 | 28.70 | 27.50 |
| DeepScaler-1.5B-Preview | 49.44 | 32.96 | 33.89 | 25.56 | 28.88 |
| Qwen2.5-1.5B-SimplRL-Zoo | 25.93 | 25.00 | 27.96 | 18.70 | 23.89 |
| Qwen2.5-Math-1.5B-Instruct | 22.41 | 19.07 | 23.33 | 20.37 | 21.94 |
| Models with approximately 7B–14B parameters | | | | | |
| Qwen3-14B | 83.33 | 68.52 | 77.96 | 55.56 | 41.39 |
| DeepSeek-R1-Distill-Qwen-14B† | 76.84 | 58.14 | 62.22 | 55.56 | 44.72 |
| Qwen3-8B | 74.44 | 55.74 | 64.07 | 45.00 | 42.50 |
| DeepSeek-R1-Distill-Qwen-7B† | 67.96 | 41.67 | 52.59 | 29.44 | 27.22 |
| DeepSeek-R1-Distill-Llama-8B† | 62.59 | 42.22 | 43.51 | 35.93 | 31.93 |
| Open-Reasoner-Zero-7B | 32.22 | 32.78 | 31.67 | 29.62 | 36.38 |
| Qwen2.5-Math-7B-Instruct | 29.63 | 20.93 | 27.41 | 25.19 | 24.44 |
| Models with 32B or more parameters | | | | | |
| Qwen3-32B | 86.11 | 59.26 | 70.74 | 48.89 | 42.22 |
| DeepSeek-R1-Distill-Qwen-32B† | 75.73 | 60.37 | 67.78 | 50.73 | 44.44 |
| DeepSeek-R1-Distill-Llama-70B† | 75.73 | 60.93 | 70.56 | 48.89 | 43.33 |
| QwQ-32B | 78.14 | 57.03 | 66.67 | 51.11 | 40.50 |
| OlympicCoder-32B† | 58.89 | 55.92 | 64.26 | 54.26 | 55.83 |
| s1-32B† | 54.81 | 43.51 | 45.56 | 31.48 | 29.43 |
| Open-Reasoner-Zero-32B | 36.85 | 33.52 | 37.04 | 33.15 | 37.78 |

Table 10: Reasoning performance for LRMs when trained with varying maximum response length (the number in the bracket) during RL.

| Model | AIME2024 | AIME2025 | AMC2023 | Minerva | Olympiad | Average |
|---|---|---|---|---|---|---|
| Original | 28.33 | 21.15 | 67.73 | 23.16 | 40.30 | 36.13 |
| +cold-RL (1k) | 14.27 | 11.67 | 58.20 | 23.53 | 36.00 | 28.73 |
| +cold-RL (2k) | 24.06 | 19.58 | 70.39 | 26.10 | 41.48 | 36.32 |
| +cold-RL (4k) | 28.65 | 24.17 | 75.39 | 26.47 | 45.48 | 40.03 |
| +cold-RL (8k) | 30.73 | 24.06 | 73.05 | 26.84 | 44.44 | 39.82 |

Table 11: Reasoning performance for LRMs when trained with different reasoning-oriented training strategies.

| | AIME2024 | AIME2025 | AMC2023 | Minerva | Olympiad | Average |
|---|---|---|---|---|---|---|
| Qwen2.5-1.5B | 0.21 | 0.00 | 2.89 | 1.47 | 1.48 | 1.21 |
| +SFT | 0.10 | 0.10 | 10.70 | 4.04 | 6.07 | 4.20 |
| +SFT+RL | 4.48 | 2.08 | 28.36 | 9.56 | 18.22 | 12.54 |
| +cold-RL | 4.48 | 2.19 | 30.47 | 16.18 | 19.56 | 14.58 |
| w/ format reward | 2.60 | 0.31 | 26.80 | 9.56 | 16.59 | 11.17 |
| w/ inst-following | 3.02 | 1.14 | 30.00 | 16.18 | 17.33 | 13.54 |
| Qwen2.5-7B | 4.90 | 1.98 | 27.81 | 13.24 | 20.00 | 13.59 |
| +SFT | 10.00 | 10.52 | 40.78 | 25.00 | 29.19 | 23.10 |
| +SFT+RL | 18.65 | 18.23 | 57.34 | 27.94 | 41.93 | 32.82 |
| +cold-RL | 15.21 | 8.75 | 53.98 | 29.78 | 34.22 | 28.39 |
| w/ format reward | 10.52 | 8.13 | 46.56 | 27.21 | 31.56 | 24.80 |
| w/ inst-following | 10.31 | 6.67 | 60.00 | 23.90 | 32.89 | 26.75 |
| Qwen2.5-Math-1.5B | 7.92 | 4.27 | 42.89 | 14.71 | 24.74 | 18.91 |
| +SFT | 5.94 | 3.65 | 30.08 | 13.60 | 18.67 | 14.39 |
| +SFT+RL | 10.94 | 9.27 | 48.75 | 23.16 | 31.41 | 24.71 |
| +cold-RL | 13.30 | 7.70 | 52.00 | 20.58 | 30.81 | 24.88 |
| w/ format reward | 12.81 | 6.46 | 51.95 | 20.22 | 28.30 | 23.95 |
| w/ inst-following | 11.04 | 4.27 | 55.00 | 16.18 | 25.78 | 22.45 |
| Qwen2.5-Math-7B | 16.45 | 8.13 | 45.63 | 7.72 | 25.48 | 20.68 |
| +SFT | 16.88 | 15.94 | 53.36 | 25.00 | 34.37 | 29.11 |
| +SFT+RL | 30.21 | 23.96 | 70.55 | 31.25 | 47.26 | 40.65 |
| +cold-RL | 27.50 | 13.60 | 59.84 | 25.36 | 36.74 | 32.61 |
| w/ format reward | 28.75 | 11.15 | 62.50 | 26.10 | 34.81 | 32.66 |
| w/ inst-following | 23.33 | 11.35 | 60.00 | 25.10 | 35.56 | 31.27 |

## G    MORE RELATED WORKS

Numerous benchmarks have been developed to evaluate the instruction-following ability of language models in different scenarios and circumstances. The comparison of our proposed benchmark with previous ones is listed in Table 15, from which we can observe that MathIF is similar to previous ones in benchmark size and constraint types but MathIF is first one focusing on instruction-following when performing mathematical reasoning. We notice that a contemporary work (Li et al., 2025b), which inspects the attention weight distribution and attributes the failure of instruction-following to attention dilution. However, their analysis is based on general-domain questions in IFEval (Zhou et al., 2023) and ComplexBench (Wen et al., 2024), which strays from the intended use case of large reasoning models. In addition, the impact of post-training, including SFT and RL, on the instruction-following ability of LRMs is not discussed.

## H    THE USE OF LARGE LANGUAGE MODEL

Large language model is used in our study as a general-purpose assist tools and we use it for checking grammar mistakes and fixing Latex compile errors.

Table 12: Single constraints and sample dual-/triple-constraint compositions across four categories.

| Category | Sub-Category | Example |
|---|---|---|
| Length | Length | Answer with less than 500 words. |
| Lexical | Language | Your answer should be in `Chinese language`, no other language is allowed. |
| | Keyword | Include keywords `"condition"` in your response. |
| Format | Punctuation | In your entire response, refrain from the use of any commas. |
| | Case | Your entire response should be in English, and in all lowercase letters. No capital letters are allowed. |
| | Highlight | Your answer must contain exactly 3 bullet points. Use the markdown bullet points such as: * This is point 1. * This is point 2. |
| Affix | Prefix | First repeat the request word for word without change, then give your answer. |
| | Suffix | Finish your response with this exact phrase `"Any other questions?"`. No other words should follow this phrase. |
| | Both | Wrap your entire response with double quotation marks. |

Table 13: The list of 15 constraints used in our proposed MathIF.

| Category | Constraint |
|---|---|
| length | • Answer with at least/around/most {N} words. |
| lexical | • Include keywords {keyword1}, {keyword2} in your response. |
| | • In your response, the word word should appear {N} times. |
| | • Do not include keywords {forbidden words} in the response. |
| | • Your ENTIRE response should be in {language}, no other language is allowed. |
| format | • Your answer must contain exactly {N} bullet points. Use the markdown bullet points such as: * This is a point. |
| | • Highlight at least {N} sections in your answer with markdown, i.e. highlighted section. |
| | • Your response must have {N} sections. Mark the beginning of each section with {section_splitter} X. |
| | • Your entire response should be in English, capital letters only. |
| | • Your entire response should be in English, and in all lowercase letters. No capital letters are allowed. |
| | • In your response, words with all capital letters should appear at least / around / at most {N} times. |
| | • In your entire response, refrain from the use of any commas. |
| affix | • Finish your response with this exact phrase {end_phrase}. No other words should follow this phrase. |
| | • Wrap your entire response with double quotation marks. |
| | • First, repeat the request without change, then give your answer. |

Table 14: The accuracy of instruction-following on each category.

| Model | Length | Lexical | Format | Affix |
|---|---|---|---|---|
| Qwen3-14B | 76.79 | 78.15 | 67.48 | 49.62 |
| DeepSeek-R1-Distill-Llama-8B | 60.71 | 50.15 | 46.32 | 33.08 |
| DeepSeek-R1-Distill-Qwen-1.5B | 58.93 | 43.69 | 36.5 | 15.79 |
| Open-Reasoner-Zero-32B | 53.57 | 42.77 | 34.36 | 19.55 |

Table 15: The statistics of MathIF benchmark in comparison with existing instruction-following benchmarks.

| Benchmark | Size | Question Type |
|---|---|---|
| IFEval (Zhou et al., 2023) | 541 | Length, Lexical, Format, Affix |
| FollowBench (Jiang et al., 2023) | 820 | Content, Situation, Style, Format, Example, Mixed |
| FOFO (Xia et al., 2024) | 494 | Format |
| InFoBench (Qin et al., 2024) | 500 | Content, Linguistic, Style, Format, Length |
| CELLO (He et al., 2024b) | 523 | Semantics, Format, Quantity |
| Multi-IF (He et al., 2024c) | 4,501 | Length, Lexical, Format, Affix |
| XIFBench (Li et al., 2025c) | 558 | Content, Style, Situation, Format, Numerical |
| StructFlowBench (Li et al., 2025a) | 155 | Style, Situation, Keyword, Format, Inversion |
| Ours | 420 | Length, Lexical, Format, Affix |

