# OpenReview forum: "Scaling Reasoning, Losing Control: Evaluating Instruction Following in Large Reasoning Models"
_ICLR.cc/2026/Conference — ICLR 2026 Conference Withdrawn Submission_

### Official Review · Reviewer_JtxE · 2025-10-25

**Soundness:** 2
**Presentation:** 3
**Contribution:** 2
**Rating:** 2
**Confidence:** 5

**Summary:**

The paper examines the pressure between reasoning ability and controllability in Large Reasoning Models. The authors introduce MathIF, a benchmark designed to assess instruction-following in mathematical reasoning contexts. Testing 23 models across scales, they find that improvements in reasoning achieved through methods such as extended CoT supervision or reinforcement learning often lead to poorer adherence to user constraints. Longer CoTs and reasoning-oriented training degrade instruction compliance, while enforcing brevity or re-emphasising constraints can restore obedience at the expense of reasoning accuracy. The study highlights an intrinsic trade-off between intelligence and obedience, underscoring the need for training paradigms that balance high reasoning competence with reliable controllability.

**Strengths:**

- novel evaluation and benchmark: MathIF is a nice systematic and domain-specific framework for measuring instruction adherence in mathematical reasoning tasks.

- good evaluation: 23 LRMs, offering a robust empirical foundation across model sizes and architectures.

- useful study: It identifies and quantifies the reasoning adherence trade-off, demonstrating that scaling reasoning capabilities often undermines instruction-following.

**Weaknesses:**

- Domain limitation: MathIF is confined to the mathematical domain; results may not generalise to broader reasoning contexts, for instance, commonsense or multimodal reasoning.

- limited training scope: The study primarily evaluates models trained via GRPO-based RL, limiting insights into other training paradigms.

- mitigation strategies: The interventions (repeating constraints) offer partial and ad hoc solutions rather than principled training methods.

- interpretability analysis missing: The work identifies the trade-off empirically but does not deeply explore cognitive or architectural causes behind the loss of control.

**Questions:**

1. Beyond empirically confirming a known trade-off, what new theoretical understanding does your study provide about why reasoning scaling reduces controllability?

2. Given that MathIF focuses exclusively on mathematical reasoning, how can you support claims of generalisability to other reasoning domains or real-world alignment scenarios?

3. How can you be sure that reasoning-oriented training, rather than factors like data or decoding strategy, causes the loss of instruction adherence? Have you tested this through controlled ablations?

---

### Official Review · Reviewer_MkMN · 2025-10-27

**Soundness:** 2
**Presentation:** 3
**Contribution:** 1
**Rating:** 4
**Confidence:** 4

**Summary:**

This paper introduces MathIF, an instruction-following (IF) evaluation benchmark for reasoning models that isolates IF performance from domain mismatch. Authors show that as models scale reasoning capacity, their obedience to instructions degrades. (When CoT length gets longer, this tendency gets stronger.) As a result, they argue an intelligence-obedience trade-off, which gains in reasoning come at the cost of controllability.

**Strengths:**

1. The paper is easy to follow, and authors provide comprehensive experiments and analyses to support their claim.
2. MathIF is the first IF benchmark particularly targeted for reasoning models that isolate IF ability from domain mismatch. The setups are well-curated and sound.
3. Authors use 25 reasoning models, which provide sound experimental supports, and further analysis in Section 5 reconfirms the limitation in training methods for reasoning models.

**Weaknesses:**

1. My biggest concern is the lack of novelty. The underperformance of instruction following of reasoning models is a well-known phenomenon in the community, and we could already observe this phenomenon using the existing benchmarks as authors provide at the first paragraph of Section 3. Leaving that aside, MathIF is an IF benchmark for math domain, where most of the settings follow the existing benchmarks. If authors want to truly isolate domain mismatch, they should conduct experiments on other domains (e.g., coding, logic, etc.). Providing single evidence in math domain and generalize into all domains feels like an over-generalization.
2. Is IF for reasoning models an important question to ask? (Can't we simply re-format the output of the reasoning models to better IF?) I (partially) agree with this, but if we assume it is important, the natural research question could be: how can we improve IF ability of reasoning models? . If this is possible, then the argued intelligence-obedience trade-off could be resolved (although authors conduct comprehensive experiments in Section 5).
3. In Section 1, I expect authors to emphasize the difference with the existing benchmark, which is isolating IF from domain mismatch. I was finally convinced while reading Section 3 first paragraph.

I'm expecting authors to persuade me during the rebuttal period. If the justifications are reasonable and persuasive, then I'm happy to raise my score.

**Questions:**

1. line 62-65: Why does longer CoT benefits reasoning performance but degrades instruction-following ability?

---

### Official Review · Reviewer_xNg7 · 2025-10-29

**Soundness:** 2
**Presentation:** 4
**Contribution:** 2
**Rating:** 2
**Confidence:** 5

**Summary:**

**Summary**: This paper analyzes how increasing a language model’s reasoning ability often leads to longer responses and reduced controllability.

Previous works that evaluate instruction-following abilities mostly focus on general-purpose tasks - simple instructions that do not require extended chains of thought. In contrast, this paper explores how reasoning-oriented training can negatively affect a model’s ability to follow user instructions.

To study this phenomenon, the authors design tasks with dual or triple constraints, ensuring that these constraints are non-contradictory through manual inspection. They introduce two evaluation metrics:

- Hard Accuracy measures whether all constraints are satisfied.

- Soft Accuracy measures whether a subset of the constraints is satisfied.

Overall, the study highlights a trade-off between reasoning strength and instruction adherence, emphasizing the need for models that are both reasoning-capable and instruction-aware.

**Strengths:**

The main strength of this paper lies in its introduction of **MathIF**, a reasoning-oriented instruction-following dataset specifically designed for the mathematical domain. This benchmark extends existing instruction-following datasets by focusing on tasks that require complex reasoning chains rather than simple directive compliance. Moreover, it introduces meaningful evaluation criteria, such as hard and soft accuracy, that better capture how well models can balance reasoning performance with instruction adherence.

**Weaknesses:**

### **1. Unclear and inconsistent reasoning - base model pairing**
The selection of base and reasoning-enhanced models is questionable. For example, DS-R1-distill-LLaMA is not a direct counterpart of Llama-3.3-70B-Instruct, as it is a distilled model derived from DeepSeek-R1. Other well-established base–reasoning model pairs such as Gemini–Gemini-1.5-Pro, Claude–Claude 3 Opus, or GPT-4o–GPT-4o-mini are ignored. This inconsistent model pairing undermines the validity of the comparisons and weakens the empirical foundation of the paper’s conclusions.

### **2. Limited novelty of the proposed benchmark**
The decline in instruction-following performance for reasoning-oriented models has already been demonstrated in prior benchmarks such as IFEval and FollowBench. The proposed MathIF dataset is thus largely an **incremental extension** rather than a fundamentally new contribution. It reuses the same conceptual framing - testing adherence to surface-level instructions, introducing deeper methodological or theoretical advances.

### **3. Misalignment between claimed goal and actual design**
Although the paper claims to focus on mathematical reasoning, the instructions tested are **primarily lexical or formatting constraints**, such as token-length limits, structured responses (e.g., bullet points, affixes), and multilingual formatting requirements. These factors are largely irrelevant to mathematical reasoning itself and do not alter or probe the underlying reasoning process. As a result, the benchmark measures surface obedience rather than true instruction-following in reasoning contexts.

### **4. Overstated claim of a “reasoning–control trade-off”**
The authors assert that there exists a fundamental trade-off between instruction-following and mathematical reasoning ability. However, their evidence (Table 2) is based on experiments where reasoning models are prompted with artificial constraints—such as token-length caps or specific output structures—that directly interfere with the reasoning process. The observed performance drop may therefore result from prompt-induced noise or confusion, not from an intrinsic trade-off.

To rigorously support their claim, the authors should conduct controlled experiments that isolate reasoning ability from instruction adherence, holding model size, training data, and training objectives constant. Measuring how reasoning and instruction-following evolve across training steps under fixed conditions would provide a more reliable causal interpretation. As it stands, the presented results support only a correlational observation, not a causal relationship.

**Questions:**

The paper is clearly written, so I do not have any questions regarding the details of the paper.

---

### Official Review · Reviewer_6nnw · 2025-11-03

**Soundness:** 3
**Presentation:** 3
**Contribution:** 3
**Rating:** 8
**Confidence:** 4

**Summary:**

The paper investigates a persistent reasoning-control trade-off in large reasoning models (LRMs). The authors introduce MathIF, a math-domain benchmark with 15 Python-verifiable constraints (namely, length, lexical, format, affix) composed into instruction sets and embedded in problems from GSM8K, MATH-500, Minerva, Olympiad, and AIME. Evaluating LRMs, they find: (i) instruction-following is low overall wrt to the target feature sets, (ii) controllability worsens with problem difficulty and constraint complexity, and (iii) model size doesn’t predict obedience. Training that boosts reasoning (SFT on long CoTs, RL variants, longer rollouts), do often erode compliance, while inference-time fixes that ‘bring constraints closer’ (e.g. repeating constraints near the answer) improve obedience but reduce correctness.

**Strengths:**

- Well designed testbed, combining different inference task types with deterministic control features.

- Comprehensive and systematic empirical analysis and interpretation.

- Clear empirical signal across different base models and tasks.

- Target properties of practical relevance.

**Weaknesses:**

- Provides further systematic corroboratory evidence to a previously known phenomena, but does not bring light to the deeper mechanisms.

**Questions:**

- Do MathIF’s constraint categories capture the instruction types real users care about (beyond format/lexical)? Any plans for pragmatic/semantic constraints with objective scoring?

- Can you isolate whether obedience loss is due to attention drift, prompt interference, or decoding heuristics (e.g., length bias)? Any controlled reordering/position ablations?

- Under equal token budgets and matched prompts, how do instruction-tuned models fare versus reasoning-RL models on MathIF?

- How sensitive are results to prompt style, delimiter choices, or different verifiers?

- ‘Intelligence’ feels like an anthropomorphic impulse here. Would stick to CoT reasoning.

---

### Note · Authors · 2026-01-06

I have read and agree with the venue's withdrawal policy on behalf of myself and my co-authors.